# Coronatine-Based Gene Expression Changes Impart Partial Resistance to Fall Armyworm (*Spodoptera frugiperda*) in Seedling Maize

**DOI:** 10.3390/genes14030735

**Published:** 2023-03-16

**Authors:** Yuxuan Lou, Xiaoxiao Jin, Zhiguo Jia, Yuqi Sun, Yiming Xu, Zihan Liu, Shuqian Tan, Fei Yi, Liusheng Duan

**Affiliations:** 1Engineering Research Center of Plant Growth Regulator, Ministry of Education & College of Agronomy and Biotechnology, China Agricultural University, Beijing 100193, China; 2MOA Key Lab of Pest Monitoring and Green Management, Department of Entomology, College of Plant Protection, China Agricultural University, Beijing 100193, China; 3College of Plant Science and Technology, Beijing University of Agriculture, Beijing 100193, China

**Keywords:** coronatine, maize, *Spodoptera frugiperda*, RNA-seq, insect resistance mechanism

## Abstract

In recent years, *Spodoptera frugiperda* (*S. frugiperda*, Smith) has invaded China, seriously threatening maize production. To explore an effective method to curb the further expansion of the harm of the *S*. *frugiperda*, this experiment used maize seedlings of the Zhengdan 958 three-leaf stage (V3) of maize as the material to study the effect of coronatine (COR) on the ability of maize to resist insects (*S*. *frugiperda*) at the seedling stage. The results showed that when maize was sprayed with 0.05 μM COR, the newly incubated larvae of *S*. *frugiperda* had the least leaf feeding. It was found that 0.05 μM COR significantly increased the contents of abscisic acid (ABA) and jasmonate (JA) in maize leaves through the determination of hormone content. Moreover, transcriptome sequencing revealed that the expression of six genes (*ZmBX1*, *ZmBX2*, *ZmBX3*, *ZmBX4*, *ZmBX5* and *ZmBX6*), which are associated with the synthesis of benzoxazinoid, were upregulated. Nine genes (*ZmZIM3*, *ZmZIM4*, *ZmZIM10*, *ZmZIM13*, *ZmZIM18*, *ZmZIM23*, *ZmZIM27*, *ZmZIM28* and *ZmZIM38*) of JA-Zim Domain (JAZ) protein in the JA signal pathway, and seven genes (*ZmPRH19*, *ZmPRH18*, *Zm00001d024732*, *Zm00001d034109*, *Zm00001d026269*, *Zm00001d028574* and *Zm00001d013220*) of ABA downstream response protein Group A Type 2C Protein Phosphatase (PP2C) were downregulated. These results demonstrated that COR could induce anti-insect factors and significantly improve insect resistance in seedling maize, which laid a theoretical foundation for further study of the mechanism of COR improving insect resistance in seedling maize, and provided data references for the use of COR as an environmentally friendly pest control method.

## 1. Introduction

*S. frugiperda* is a global migratory agricultural pest that causes damage to a wide range of crops, spreads swiftly, and has a broad impact [1,2,3,4]. It seriously threatens maize production [5,6,7]. *S. frugiperda* has migrated to China from Myanmar and spread to 26 provinces (municipalities and autonomous regions) [8,9]. The unexpected invasion and fast spread of *S. frugiperda* has posed significant challenges to preventative and control efforts [10]. To control *S. frugiperda*, a variety of pesticides such as organophosphorus, carbamates, pyrethroids, avermectins, amides, spinosad, and others are utilized [11]. However, there are food safety and ecological safety problems in the use of chemical pesticides, and the field populations of *S. frugiperda* have varying degrees of resistance to chemicals [12]. Therefore, it is urgent to develop new agents and methods to control *S. frugiperda*.

COR is a phytotoxin generated by *Pseudomonas syingae* pathovars [13,14,15], with a similar function as JA [16,17]. It has been demonstrated that COR is an analog of JA, and is 1000 times more active [18,19]. Recent studies have shown that COR can regulate maize morphogenesis and promote maize growth at the seedling stage [20]. It can also improve lodging resistance and enhance maize resistance to low temperature and drought [21,22,23]. However, it has not been reported that COR can improve pest resistance of maize at the seedling stage. Studies have shown that when plants suffer from pests, elicitors and injury signals, they promote the synthesis of JA [24]. JA receptor COR insensitive 1 (COI1) can mediate the opening of the plant defense response [25] and COR is an effective agonist of COI1. Therefore, we hypothesize that COR treatment may activate the downstream signal transduction pathway of the elicitor and induce the synthesis of insect-resistant metabolites and stress-resistant plant hormones, which is like that of the elicitor itself.

The role of COR in improving maize insect resistance at the seedling stage was studied by using chemical regulation. This study focused on the biosynthesis of plant hormones and other insect-resistant secondary metabolites, especially JA and benzoxazines. This study was done to find out how COR makes maize resistant to *S*. *frugiperda* and how it works on a molecular level. This would give a basis for applying the plant growth regulator to maize production to stop pests.

## 2. Materials and Methods

### 2.1. Plant and Insect Materials

The hybrids of maize ZD958 used in our experiment were collected from the Henan Golddoctor Seeds Co., Ltd., which were cultivated in Beijing (40°10′ N, 116°20′ E), China, during April of 2022. The maize was cultivated in the greenhouse with 25 °C/18 °C day/night temperature and 16 h/8 h photoperiod. The eggs of *S. frugiperda* were collected from the Henan Jiyuan Baiyun Industry Co., Ltd. (Jiyuan, China).

### 2.2. The Method of COR Treatment

COR was purified by the Centre for Crop Chemical Control and the purity was >99%, measured with high-performance liquid chromatography (Milford, MA, USA). COR was diluted to different concentrations by water before foliar spraying. The time of spraying COR was the third day after the 3rd leaf was fully deployed. At this time, the maize was at the seedling stage (about 21 days after sowing) and the third and fourth leaves did not easily lose water and deform, so it was convenient to measure the notch area of the maize leaves later. According to the work of He et al., after spraying, COR will enter plant tissues and be transported from shoot to root, and the COR content of maize leaves and stems reached the highest value at 24 h after the application of COR [26]. The pattern of COR distribution in different maize seedling tissues can be found in ‘Appendix A’ from He’s article. The total amount of liquid was 2 mL·plant^−1^. The maize treated with the same amount of water was the control. Tween was added at 0.5‰ to promote the absorption of the solution by maize leaves.

### 2.3. Determination of the Feeding Amount of the Newly Hatched Larvae of S. frugiperda on the Leaves of Maize Sprayed with COR

Five experimental groups were set up for COR treatment with maize, viz., 0.01 μM, 0.05 μM, 0.1 μM, 1 μM and 2 μM, and 1 control group was set up for water treatment. After 24 h of COR treatment, three leaves with the same growth (the fourth) from each group were washed with clean water to exclude the effect of COR odor and fed ten newly hatched larvae of *S. frugiperda* (divided into three groups of biological repetitions). To prevent the larvae from running out, we sealed the petri dish containing the maize leaves and larvae with a sealing film. After 24 h, the engraving area of the maize leaves were photographed and measured using image J and compared between groups. Considering the effect of water loss, the original engraving area was reduced according to the wilting rate. Mathematical statistics and drawing were done through the Windows version of GraphPad Prism (V7.0) (San Diego, CA, USA).

### 2.4. Determination of Endogenous Plant Hormones in Maize Leaves

There were two treatments: 24 h after treatments of the 0.05 μM COR and water treatment group, twelve leaves with similar growth (the third and fourth leaves) were taken from each of the two groups, and the contents of endogenous plant hormones in the 0.05 μM COR treatment group and water treatment group were measured by enzyme-linked immunosorbent assay (ELISA) [27]. They were divided into three groups of biological repeats with four leaves each. The antigens, antibodies, hormone standards and horseradish peroxidase (HRP)-labelled secondary antibodies were provided by the Crop Chemical Control Research Center of China Agricultural University.

### 2.5. RNA Extraction and Library Preparing

RNA extraction and library preparing proceeded according to the method of Ren et al. [21]. Total RNA was extracted using Trizol (Invitrogen, Carlsbad, CA, USA) based on the manual. Then it was purified by magnetic stand (Invitrogen, Carlsbad, USA). The purified total RNA was stored in a −80 °C freezer. The libraries of sequencing were constructed following the instructions of the manufacturer.

The library construction proceeded accorded to the protocol of Illumina, and synthetic cDNA was treated with end-repair and phosphorylation. PCR for 15 cycles using NEB’s Phusion DNA polymerase and selection of size was performed for target fragments of cDNA on 2% agarose. All libraries of paired-end sequencing were sequenced using the HiSeq xten.

### 2.6. RNA-Seq Data Analysis

To align the paired-end readings and ensure the quality of readings, we trimmed the readings of paired ends with the SeqPrep (https://github.com/jstjohn/SeqPrep, accessed on 22 May 2022) and filtered the illumina readings with the Sickle (https://github.com/najoshi/sickle, accessed on day month year.). Then, the mapping of readings to the reference genome was performed using the Hisat2 [28]. The unique mapped readings were processed using Cufflinks (V2.2.0) (San Diego, CA, USA) [29]. Fragments per kilobase of transcript per million mapped reads (FPKM) was used to indicate the gene expression level. The genes whose FPKM values were greater than 1 were selected, and then the scatter plot was drawn and the *R*^2^ between biological replicates was calculated by the Pearson algorithm with omicshare (https://www.omicshare.com, accessed on 22 May 2022).

Three biological repeats were set for each treatment (water and 0.05 μM COR). Each replicate was obtained by pooling samples from at least three plants. To ensure the reliability of the analysis results, the genes with FPKM values > 0 in all the six samples were chosen to draw the correlation map (Appendix A). According to the correlation values between each two biological repeats, we eliminated a set of data in each treatment (control3, COR3).

In the Lianchuan biological cloud platform (https://www.omicstudio.cn., accessed on 25 May 2022), the genes with FPKM values > 0 in the remaining four samples were selected for PCA analysis and cluster analysis (the data were processed by log_2_ and normalized by Z-score).

### 2.7. The Analysis of Differential Expression

Each transcript’s expression level was indicated by FPKM value. Then the differentially expressed genes were calculated by Cuffdiff (http://cufflinks.cbcb.umd.edu/, accessed on 22 May 2022) [30].

### 2.8. Functional Enrichment Analysis

All the differentially expressed genes were chosen to have a KEGG enrichment analysis by David [31].

## 3. Results

### 3.1. Insect Resistance of COR with Different Concentrations

Experimental-group maize seedlings of Zhengdan 958 three-leaf stage (V3) were treated with COR of 0.01 μM, 0.05 μM, 0.1 μM, 1 μM and 2 μM and control group maize leaves were treated with water. After one day, the newly hatched larvae of *S*. *frugiperda* were fed with the leaves. It was found that the leaf notch area treated with 0.05 μM COR for 24 h was significantly different from that of the control group (Figure 1b), and the notch area per unit time per unit count of insects decreased by 67.8% compared with the control group, which was the smallest of all the groups. The notch area per unit time per unit count of insects of other concentration treatment groups was also decreased, but there was no significant difference between them and the control group (Figure 1c). The results showed that the treatment of 0.05 μM COR could significantly improve the insect resistance of maize at seedling stage and prevent the decrease of photosynthetic effective area caused by insect feeding.

### 3.2. Changes in Endogenous Plant Hormones

To clarify the mechanism of COR improving the insect resistance of maize at the seedling stage, we determined the contents of five endogenous plant hormones, ABA, JA, indole acetic acid (IAA), gibberellic acid (GA) and zeatin riboside (ZR). After spraying 0.05 μM COR or water for one day, the contents of ABA and JA in maize leaves in the treatment group were significantly higher than those in the control group, increasing by 461.8% and 44.3% respectively (Figure 2a,b), while the levels of other hormones did not change (Appendix A). This indicates that the ABA and JA may relate to the improvement of insect resistance of maize.

### 3.3. Overall Analysis of Transcriptome in Response to COR Treatment

To further analyze the molecular mechanism of COR improving the insect resistance of maize seedlings, we obtained the transcriptome data of maize seedlings treated with 0.05 μM COR and water for 24 h. There were two biological repeats for each treatment, each of which came from six leaves (the third leaf and the fourth leaf) from three different plants. In total, 189 million reads were generated on the illumina sequencing platform, and then mapped to the maize B73 reference genome [32]. An average 81.48% of reads were mapped and an average 55.02% of reads were mapped uniquely (Table 1). Then, the uniquely mapped reads were further used to calculate the normalized gene expression level as FPKM. The data of four samples were analyzed by principal component analysis (PCA) and cluster analysis, and the results showed that the difference between groups was greater than that within groups (Figure 3a,b). The comparison of two biological replicates showed that their FPKM values were highly correlated (average *R*^2^ = 0.963, Figure 3c,d). Therefore, we used the average FPKM value as the expression level of the treatment and control. In order to reduce the impact of transcription noise, it was only when the FPKM value of a gene was greater than 1 that we considered it to be expressed. In total, 18,741 genes including 1371 transcription factors (TFs) expressed in at least one of the groups. The above conclusions show that the results of the transcriptome sequencing are sufficient to support further data analysis.

### 3.4. Identification of Differentially Expressed Genes after COR Treatment

We took fold change larger than 2 or less than 1/2, expression in at least one sample and q-value smaller than 0.05 as the criteria for screening differentially expressed genes. Compared with the control group, there were 170 upregulated genes and 734 downregulated genes after 0.05 μM COR treatment (Figure 4a, Data sets 1). KEGG enrichment analysis showed that the differentially expressed genes were significantly enriched into eight categories (*p* < 0.05) (Figure 4c). The biosynthesis of secondary metabolites (zma01110), the benzoxazinoid biosynthesis (zma00402), which plays role in plant defense, and the plant hormone signal transduction (zma04075), which relates to the response to biotic and abiotic stress [33] were significantly enriched (Appendix A), indicating that the mechanism of COR improving the insect resistance of maize seedlings is related to insect-resistant substances and plant hormones.

### 3.5. Expression of Genes Related to the Biosynthesis of Benzoxazines

To further understand the correlation between the metabolism of insect-resistant substances and the enhancement of the insect resistance of maize seedlings, we paid attention to the differential expression of genes related to the biosynthesis of benzoxazines, which are the most important secondary defense metabolites in Gramineae [34,35,36,37]. After spraying 0.05 μM COR, the expression of *ZmBX1*, *ZmBX2*, *ZmBX3*, *ZmBX4*, *ZmBX5* and *ZmBX6* (Appendix A) related to the synthesis of benzoxazines was upregulated by 63.7%, 455.2%, 89.8%, 77.9%, 122.2% and 43.2%, respectively (Figure 5). It is suggested that the synthesis of anti-insect active substance benzoxazines is one of the potential mechanisms of COR to enhance the insect resistance of maize at the seedling stage.

### 3.6. Expression of Genes Related to JA and ABA in Maize

In order to explore the role of plant hormones in improving the insect resistance of maize at seedling stage, we analyzed the differentially expressed genes related to plant hormones (Appendix A). Of all the plant hormones, we found the most significant changes in the expression of genes related to JA and ABA (Appendix A).

The components of the JA signal transduction pathway, especially JAZ proteins, play an important role in plant response to elicitors [38]. ABA plays an important regulatory role in plant response to a variety of biotic and abiotic stresses [39,40]. We found that most genes related to ABA and JA biosynsynthesis and anabolic metabolism were downregulated. The nine JAZ family genes *ZmZIM3*, *ZmZIM4*, *ZmZIM10*, *ZmZIM13*, *ZmZIM18*, *ZmZIM23*, *ZmZIM27*, *ZmZIM28* and *ZmZIM38* encoding negative regulatory proteins of the JA signal pathway were downregulated by 74.0%, 66.1%, 53.8%, 69.8%, 79.6%, 59.7%, 76.8%, 60.1% and 59.0%, respectively, and five genes of the MYC2 family, *ZmMYC7*, *ZmBHLH108*, *ZmBHLH116*, and *ZmBHLH91* and *ZmBHLH57* were downregulated by 59.5%, 87.8%, 90.9%, 67.4% and 77.8%, respectively. (Figure 6a). The genes *ZmPRH19*, *ZmPRH18*, *Zm00001d024732*, *Zm00001d034109*, *Zm00001d026269*, *Zm00001d028574* and *Zm00001d013220* of seven negative regulatory proteins encoding ABA signal transduction group A group 2C protein phosphatase (PP2C) were significantly downregulated by 67.4%, 64.1%, 55.9%, 56.3%, 56.0%, 73.8%, 62.5%, respectively, (Figure 6b), indicating that COR can improve the insect resistance of maize at the seedling stage through ABA and JA signal pathways.

## 4. Discussion

Plants have developed internal systems to respond to adversity in order to adapt to stress. Plants may withstand herbivore invasions, such as those caused by insects, in the wild by perceiving and reacting to elicitors [41]. The JA signaling pathway components are critical in a plant’s response to elicitors. COR has substantial scientific significance to knowledge about the mechanism that plants induce for insect resistance, since it can bind to the JA receptor COI1 and has stronger biological activity than JA, which may help in developing crop resistance to insects. However, it is worth noting that coronatine can promote pseudomonas virulence by inhibiting salicylic acid accumulation, and spraying for insect resistance will make it susceptible to Pseudomonas [42]. Therefore, the application of coronatine in production needs further consideration.

### 4.1. COR can Effectively Improve the Insect Resistance of Maize at the Seedling Stage

It was observed in this study that foliar spraying of maize seedlings with 0.05 μM COR might greatly boost insect resistance. Based on the finding, we think the further research on COR may help in developing crop resistance to insects and to reduce *S*. *frugiperda* in ecologically friendly grassland.

### 4.2. The Enhanced Synthesis of Benzoxazines May Be One of the Reasons for the Improvement of the Insect Resistance of Maize at the Seedling Stage

We discovered that *ZmBX1*, *ZmBX2*, *ZmBX3*, *ZmBX4*, *ZmBX5* and *ZmBX6* genes were considerably upregulated in leaves one day after spraying with 0.05 μM COR. These genes encode proteins that catalyze the production of benzoxazines in maize. The increased synthesis of benzoxazines generated by upregulated expression may contribute to maize seedling insect resistance.

### 4.3. COR May Improve the Insect Resistance of Maize at the Seedling Stage through the ABA and JA Signal Pathways

JA plays a key role in the elicitor signal transduction pathway, and is important in plant response to insect pests [43,44,45]. Through the negative regulation of the JAZ protein, plants can enhance their stress resistance [46,47]. Moreover, JA works as a signal transducer in the induction of the accumulation of benzoxazines [48]. It was found that after treating with 0.05 μM COR for one day, the expression of nine JAZ protein genes were significantly downregulated, and the JA content in maize leaves increased significantly. This may be the reason for the enhanced anabolism of benzoxazines. In addition, the content of ABA in maize leaves related to stress response increased one day after spraying with 0.05 μM COR, while the expression of PP2Cs, an inhibitor of ABA signal pathway, decreased. These changes in hormones and signals may be an important mechanism for the enhancement of insect resistance of maize at the seedling stage induced by COR. According to the result, we proposed the possible mechanism of COR-induced insect resistance in maize (Figure 7).

## 5. Conclusions

Through transcriptional sequencing and measuring the content of stress-resistant phytohormones, this study examines the effects of various concentrations of COR on the insect resistance of maize at the seedling stage. It also discusses the mechanism by which COR improves the insect resistance of maize at the seedling stage. Theoretically, the findings support COR-based environmentally friendly pest control.

## Figures and Tables

**Figure 1 genes-14-00735-f001:**
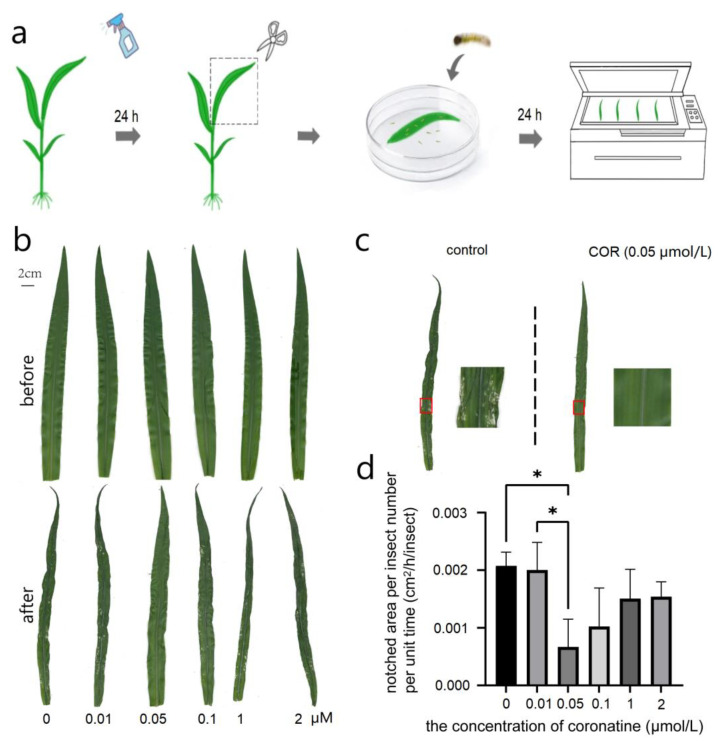
The notch area of maize leaves sprayed with different concentrations of COR per unit time per unit count of insects. (**a**) The protocol of insect feeding on maize seeding. (**b**) Comparison of the state of maize leaves fed by the newly hatched larvae of *S*. *frugiperda* for 24 h. (**c**) Comparison of control and 0.05 μM COR treatment on seedling insect feeding phenotypes. (**d**) The notch area of maize leaves per unit time per unit count of insects. The notched areas of maize leaves were photographed with blade scanner and measured using image J. The data were analyzed by one-way ANOVA and Tukey multiple comparisons. * Indicates that the difference is significant, and the significant degree α is 0.05.

**Figure 2 genes-14-00735-f002:**
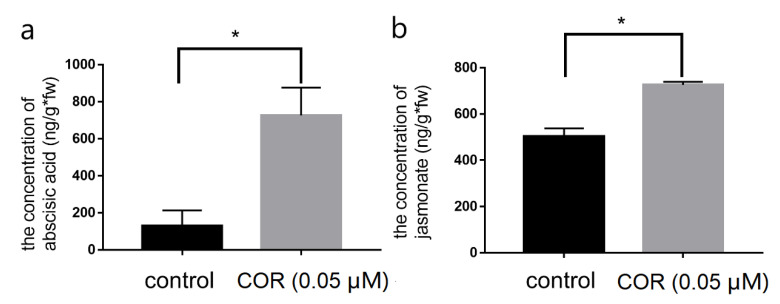
Effect of COR on endogenous hormone contents in maize seedings. (**a**,**b**) Changes of ABA (**a**) and JA (**b**) contents in maize leaves induced by 0.05 μM COR. The data were presented as means ± SE (*n* = 3) and analyzed by Student’s *t*-test. SE is represented by error bars. * Indicates that the difference is significant, and the significant degree α is 0.05.

**Figure 3 genes-14-00735-f003:**
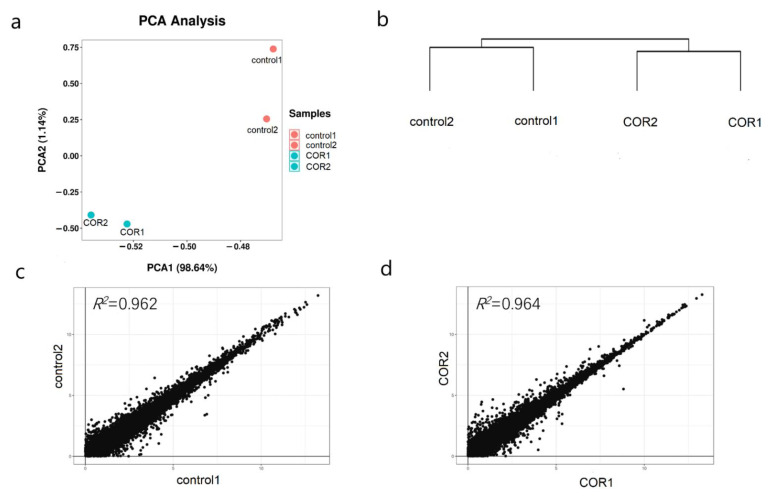
Intra-group correlation and inter-group difference. (**a**,**b**) PCA analysis and cluster analysis by complete cluster. (**c**,**d**) Take the log_2_ (FPKM) value of control1 as the abscissa, the log_2_ (FPKM) value of control2 as the ordinate, and the log_2_ (FPKM) value of COR1 as the abscissa and the log_2_ (FPKM) value of COR2 as the ordinate to draw the scatter charts and calculate the correlation coefficient.

**Figure 4 genes-14-00735-f004:**
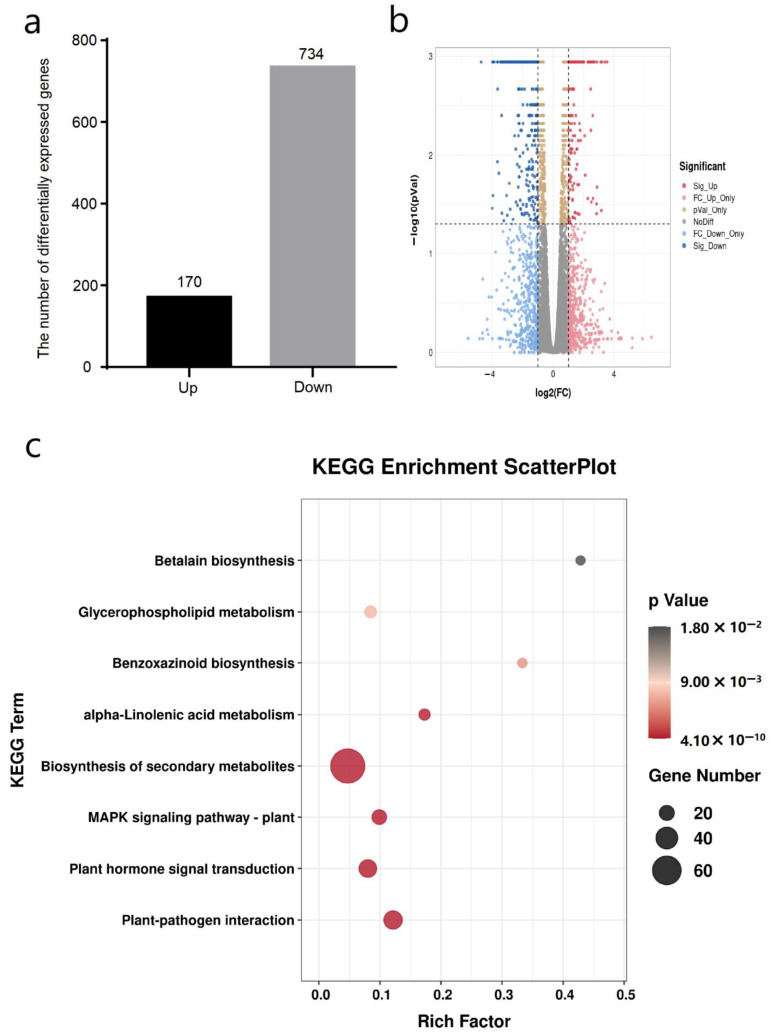
Analysis of differentially expressed genes in maize leaves induced by COR. (**a**) The number of differentially expressed genes. COR treatments gave output of 170 upregulated and 734 downregulated genes. (**b**) Volcano plot of genes’ differential expression. (Sig_ Up, FC_ Up_ Only, pVal_ Only, NoDiff, FC_ Down_ Only, Sig_ Down represent genes that are significantly upregulated, FDR > 0.05 and fold change value is greater than 2, FDR < 0.05 and fold change value is greater than 1/2 or less than 2, FDR > 0.05 and fold change value is greater than 1/2 or less than 2, FDR > 0.05 and fold change value is less than 1/2, and significantly downregulated, respectively). (**c**) KEGG enrichment analysis of differentially expressed genes.

**Figure 5 genes-14-00735-f005:**
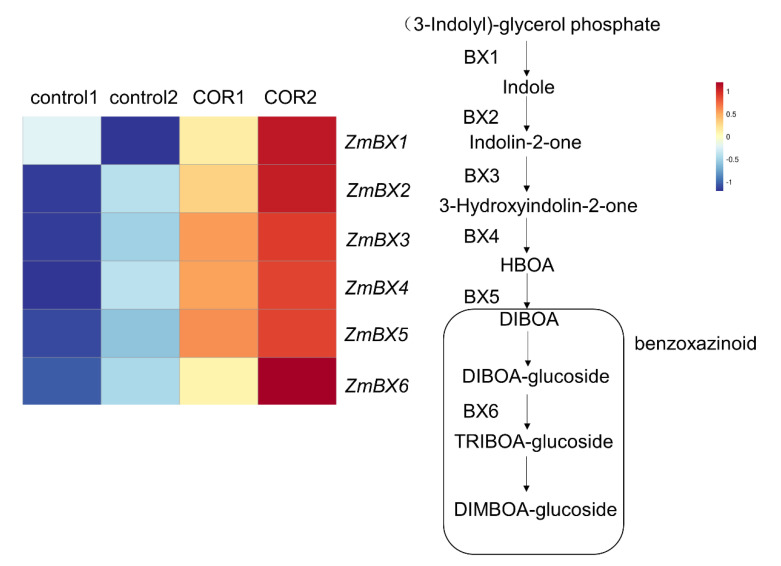
Expression of genes related to the anabolism of benzoxazines. The data were normalized by log_2_ and Z-score; the closer to 1, the higher the relative gene expression, and the closer to −1, the lower the relative gene expression. Control1 and control2 are two biological duplicates of control group and COR1, COR2 are two biological duplicates of 0.05 μM COR treatment group. The enzymatic reaction pathway is shown with arrows.

**Figure 6 genes-14-00735-f006:**
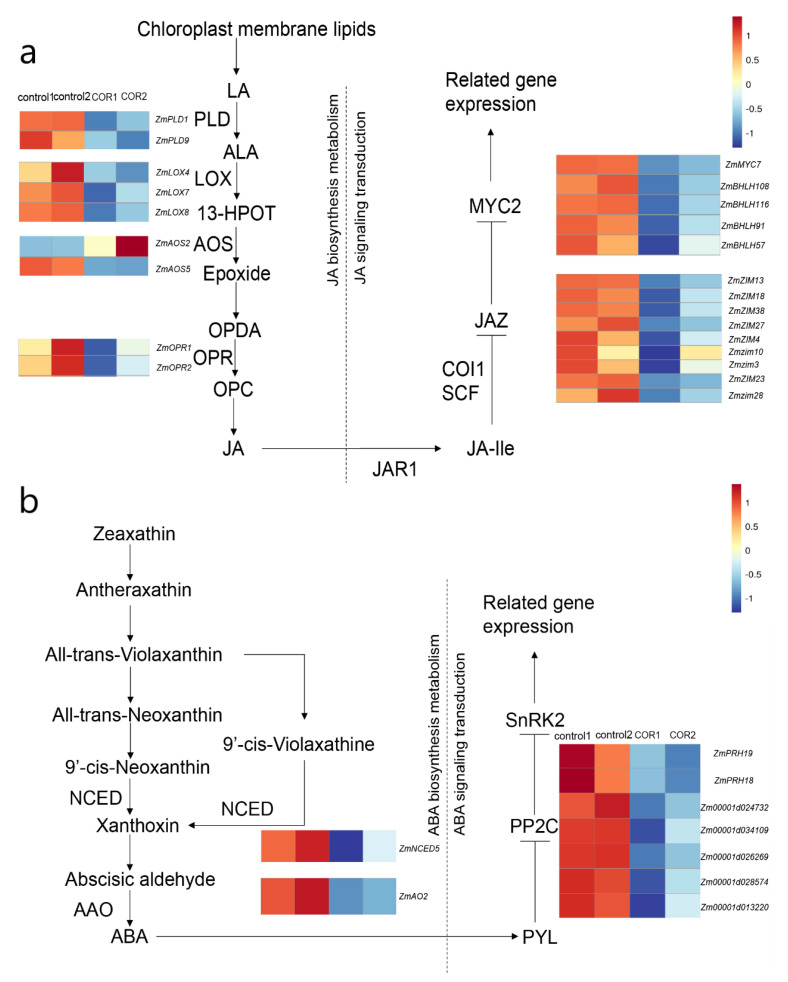
The expression of genes in JA and ABA pathways was affected by COR. (**a**,**b**) Effects of COR on JA (**a**) and ABA (**b**) pathway genes. The data were normalized by log_2_ and Z-score; the closer to 1, the higher the relative gene expression, and the closer to −1, the lower the relative gene expression. The enzymatic reaction pathway is shown with arrows.

**Figure 7 genes-14-00735-f007:**
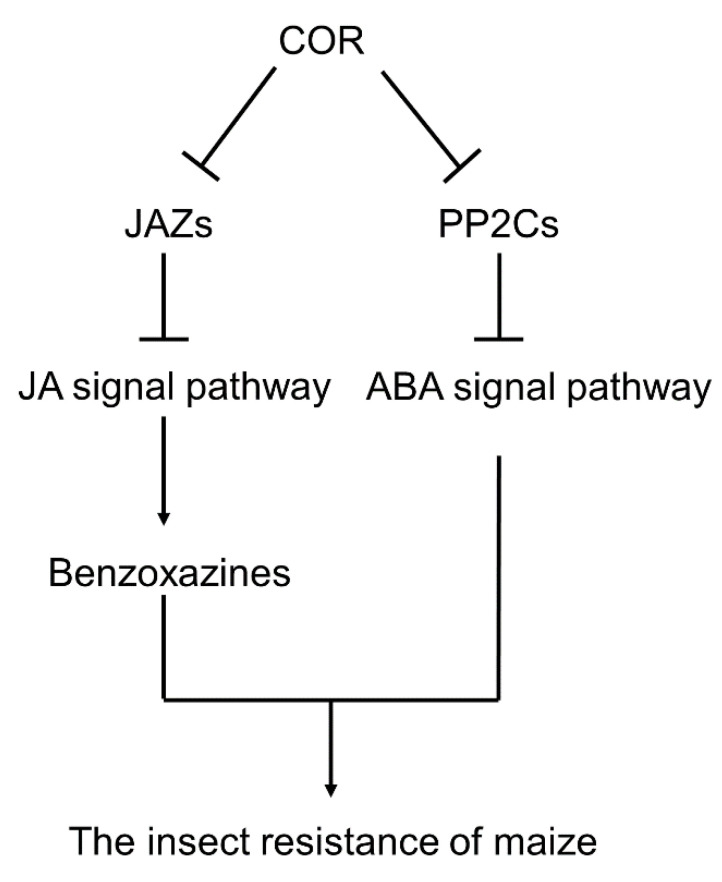
Model graph of the insect resistance of maize regulated by COR treatment.

**Table 1 genes-14-00735-t001:** RNA-seq readings: summary of mapping results.

		Mapped Readings		Unique Mapped Readings	
Sample	Raw Readings	Number	% Mapped	Number	% Mapped
control1	46,020,064	37,748,120	82.03	25,691,854	55.83
control2	46,188,984	37,766,492	81.77	25,679,736	55.68
COR1	51,153,190	41,618,084	81.36	27,964,204	54.67
COR2	45,283,680	36,566,492	80.75	24,400,408	53.88

Control1 and control2 are two biological duplicates of control group and COR1, COR2 are two biological duplicates of 0.05 μM COR treatment group.

## Data Availability

Transcriptome information from this research can be downloaded in the NCBI Sequence Read Archive (http://www.ncbi.nlm.nih.gov/sra, accessed on 20 February 2023) through accession number PRJNA926916.

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
