# Peer review of "Coronatine-Based Gene Expression Changes Impart Partial Resistance to Fall Armyworm (Spodoptera frugiperda) in Seedling Maize"

_genes, 2023, doi:10.3390/genes14030735_

Round 1

Reviewer 1 Report

A brief summary

Fall armyworm (FAW) has become global problem.  The authors of this manuscript conducted experiments in attempts finding solution in FAW management by looking at coronatine effects at seedling stage of maize.  This is a very interesting paper and it suggested FAW control tactics that are safe to the environment.  The manuscript was easy to follow and understand.  The introduction section was brief yet provided good background and overview leading to why this study was conducted.  It provided excellent reason why one would want to test coronatine (COR) in relation to FAW control.  The materials and methods detailed the methodology on how the experiments were done and could be replicated by others who would like to do similar studies. The figures presented in results section enhanced the explanation in results and discussion sections. The results from this study are beneficial for other scientists who conduct similar research efforts in combatting damage caused by FAW throughout the world.  In the discussion section the authors presented potential future use of the results of this study in control of FAW for those interested conducting further studies. 

Specific comments:

Below are a few suggestions to improve the quality of the manuscript and provide better scientific explanation:

Materials and Methods (page 2-3)

Lines 63-93 (Sections 2.1, 2.2, 2.3, 2.4)

The corn leaves used for experiments were V-3 (3rd stage) or V-4 (4th stage).  It is not stated anywhere why these stages were selected – although I assumed because the study focused on seedling stage.  A brief explanation on the selection of the stage would be helpful for readers who aren’t familiar with the subject.

Line 97 (Section 2.5)

… -80°C refrigerator. à It is uncommon to call -80°C as refrigerator.  “Refrigerator” verbiage should be changed to “freezer” or “ultrafreezer”.

Results (page 4-10)

Lines 150-157 (page 4)

Figure 1: How did the image was generated?  I understand it was mentioned in the Materials and Methods section but it should be briefly stated here.

Line 169 (page 4-5)

It seems the beginning of this line is missing words and it does not read correctly: and (b): …

Line 188 (page 5)

TF – please spell out for the first time [although there is abbreviation section at the end, it would be wise to spell out for the first time]

Line 191 (page 5)

What are control 1, control 2, COR1, COR2?  Footnote below the table to describe each one would be helpful.

Line 205 (page 6)

… in plant defence … à should be plant defense.

Line 210 (page 7)

Figure 4a: Figure legend “control vs COR” is misleading and should be revised.  The legend needs to reflect the information would like to present through the figure, i.e. COR treatments gave output of 170 up-regulated and 734 down-regulated genes.

Line 229 (page 8)

Figure 5 left picture:  What is the difference between COR1 and COR2?  Add a note next to Figure 5 title.

References (page 12-14)

Lines 337-446: Species name needs to be italicized.

Reviewer 2 Report

The MS “Coronatine induced anti-insect factors and improved insect resistance of maize at seedling stage” by Lou et al provides preliminary insight into gene expression changes in maize and will be interesting to know about the bona fide mechanism that plants induce for resistance. Most of the hormones don’t directly control insect. Rather they deploy secondary mechanism for insect control.

Major suggestions

1.      In this regard the title must be appropriately changed.

2.      Coronatine Promotes Pseudomonas Virulence by Inhibiting Salicylic Acid Accumulation. In this regard spraying for insect resistance will make it susceptible to Pseudomonas. Pl see the citation by Zheng et al 11 (6). year  2012, Pg 587-596.

3.      How the coronatine was made permeable into the plant tissue …An analysis of spraying concentration and intra cellular concentration post-spray may be provided to support the induction of gene expression.

Minor suggestions

4.      Line no. 35 & 36 prescribed dates are not required… citations are enough.

5.      Line no. 71 the methods of arriving “third day third leaf stage” may be explained

6.      Line no. 77 what is set five experimental groups… needs more explanation

7.      Line no. 90 “Endoge plant” makes no sense

8.      Line no. 95 & 96 “was accepted for”…. check for grammar. Basing on the manual may be corrected

9.      Line no. 102 “after PCR treated” is not meaningful may be corrected

10.  Line no. 109  “and control” needs to be looked at carefully for meaningful sentence.

11.  Line no. 122 check for grammar (choosed)

12.  Figure 1 A closeup showing the tunnels in leaf must be provided.

13.  Figure 1 B and line 156 significant difference must be noted by */**. What is the meaning of salience alfa

14.  Line 160 & 161 how where five endogenous plant hormones were estimated and protocol is missing in methods section.

15.  Line 175 how the authors arrived at 0.05 microMolar COR? Fig 1B shows 0.1/1.0 concentration are better than 0.5.

16.  Line no. 197 PCA analysis pl check writing as it does not make any sense here.

17.  Line no. 257 Log2 Subscript

18.  Line no. 264 “Substantial scientific significance” must be quantified

19.  Line no. 268-275 not relevant for result section as it is already reported.

20.  Line no. 277 spraying COR is a proposition and not advisable for field scale application due to economically unviable cost.

21.  Line no. 285 not relevant for the result section

22.  Line no. 294 check the grammar.

23.  Line no. 300 not a bona fide mechanism for insect resistance

24.  Line no. 303 check the syntax to make it meaningfull.

Look at the highlighted text in the attached MS pdf.

Reviewer 3 Report

1. Authors should clearly provide all the plant materials they have used in the experiment (materials and method).

2. Authors should provide a clear flowchart/Tablular presentation to understand the process of infection protocol.

3. Author should give any reference in point 2.5 (RNA extraction and preparing library)

4. Authors should give more clear picture and mark affected areas in figure 1.

5. Authors should check the typological error in legends of figure 2 marking.

6. Authors should correct the Figure numbers (a and b) are wrongly placed in figure 3. 

Round 2

Reviewer 2 Report

Pl see the attahced word document with suggstion.

The title may be changed as suggested by accepting other reviewers and EIC recommendation.
